# CROSS-SCENARIO UNIFIED MODELING OF USER INTERESTS AT BILLION SCALE

## ABSTRACT

User interests on content platforms are inherently diverse, manifesting through complex behavioral patterns across heterogeneous scenarios such as search, feed browsing, and content discovery. Traditional recommendation systems typically prioritize business metric optimization within isolated specific scenarios, neglecting cross-scenario behavioral signals and struggling to integrate advanced techniques like LLMs at billion-scale deployments, which finally limits their ability to capture holistic user interests across platform touchpoints. We propose *RED-Rec*, an LLM-enhanced hierarchical *R*ecommender *E*ngine for *D*iversified scenarios, tailored for industry-level content recommendation systems. *RED-Rec* unifies user interest representations across multiple behavioral contexts by aggregating and synthesizing actions from varied scenarios, resulting in comprehensive item and user modeling. At its core, a two-tower LLM-powered framework enables nuanced, multifaceted representations with deployment efficiency, and a scenario-aware dense mixing and querying policy effectively fuses diverse behavioral signals to capture cross-scenario user intent patterns and express fine-grained, context-specific intents during serving. We validate *RED-Rec* on hundreds of millions of users in a world-leading UGC platform through online A/B testing, showing substantial performance gains in both content recommendation and advertisement targeting tasks. We further introduce a million-scale sequential recommendation dataset for comprehensive offline evaluation. We hope our work could advance unified modeling of users, unlocking deeper personalization and fostering more meaningful user engagement across large-scale platforms.

## 1 INTRODUCTION

Modern content platforms have evolved into complex ecosystems where users engage across multiple behavioral contexts—browsing personalized feeds, conducting topical searches, discovering content creators, and responding to targeted advertisements. Each interaction scenario captures distinct yet complementary aspects of user intent: search queries reveal explicit informational needs, feed engagement demonstrates implicit content affinity, and advertisement responses indicate commercial preferences (Covington et al., 2016; Hidasi et al., 2015). As a result, user interests are inherently multi-dimensional and dynamic, manifesting through intertwined behavioral trajectories spanning these contexts (Figure 1).

Despite this richness, production recommendation systems typically operate in isolation, with separate models independently optimized for business objectives such as Click-Through Rate (CTR) in feeds and Advertiser Value (*ADVV*) in advertisements (Zhang et al., 2019; Chapelle et al., 2014). This siloed design, which traps systems in local optima, leads to several limitations. First, it fragments user understanding by restricting each model to narrow behavioral contexts, thereby weakening generalization and robustness. Second, it yields inconsistent user experiences when independent systems infer divergent interests. Third, it underutilizes cross-scenario signals, limiting knowledge transfer across tasks and weakening performance for users with sparse activity in certain scenarios (Xia et al., 2020; Zhu et al., 2022). For example, a user exploring sustainable living may search for eco-friendly products, engage with environmental content, and click green-technology ads—yet scenario-isolated modeling fails to synthesize coherent signals into unified user intent. Although some cross-scenario approaches exist (Zhang et al., 2023; Bao et al., 2023), they typically require extensive manual feature engineering and struggle with scalability and robustness in production.

Figure 1: **Learning user interests from diverse behavioral contexts.** *Left:* User interests manifest through diverse topics and interconnected behavioral patterns across multiple engagement contexts. *Middle:* These multifaceted interests are naturally captured in rich behavioral sequences spanning homefeed interactions, ad engagements, and search activities, reflecting the dynamic and evolving nature of user preferences. *Right:* RED-Rec employs unified hierarchical sequential representation learning based on LLMs to process these behavioral histories, generating nuanced user embeddings that enable context-aware recommendations. Terminology is further explained in Section A.

We are motivated by the observation that users exhibit consistent interest patterns across diverse scenarios, and that modeling these patterns holistically can significantly enhance recommendation quality. The observation highlights the value of user-centric, instead of scenario-centric, systems that synthesize behavioral signals from multiple scenarios to construct comprehensive user interest representations. Recent advances have made this approach increasingly feasible: LLMs have transformed the semantic understanding of user behaviors and content (Wang et al., 2024b), while advanced sequence modeling techniques effectively capture complex temporal dynamics and cross-scenario dependencies (Sun et al., 2019). Meanwhile, modern platforms generate massive multi-scenario logs (McAuley et al., 2015; Harper & Konstan, 2015; Gao et al., 2022), creating opportunities for unified modeling at scale. However, realizing this vision entails significant challenges: (i) heterogeneity in action schemas, temporal dynamics, and semantics; (ii) activity imbalance across scenarios; (iii) large-scale training and serving with strict latency and throughput constraints; and (iv) reconciling differing optimization objectives within a single architecture. While recent work explores mixtures of multi-source signals (Ma et al., 2022; Zhang et al., 2022a; Liu et al., 2024; Yang et al., 2024), truly end-to-end unified modeling for industrial deployments remains underexplored.

We present *R*ecommender *E*ngine for *D*iversified scenarios (*RED-Rec*), an LLM-enhanced hierarchical sequential recommendation framework tailored for billion-scale content platforms. *RED-Rec* unifies interest modeling across heterogeneous contexts by: (1) employing LLM-powered user and item encoders within a hierarchical two-tower structure, enabling rich semantic representations while preserving efficiency for large-scale retrieval; (2) introducing a 2-D dense mixing policy that fuses multi-scenario behavioral signals along temporal and scenario axes to capture cross-scenario dependencies, coupled with multi-interest, scenario-aware queries that express fine-grained, scenario-specific user intents. We train *RED-Rec* end-to-end on billions of behavioral events drawn from billions of items and over one hundred million users, and incorporate system-level optimizations enabling stable, low-latency online deployment.

To enable a more rigorous evaluation, we additionally introduce a new multi-scenario sequential dataset, curated from a world-leading User-Generated Content (UGC) platform. The dataset spans millions of items and diverse user behaviors across feeds, search, and advertisement contexts, facilitating benchmarking of unified and scenario-specific models. Through extensive offline experiments, *RED-Rec* consistently outperforms baselines across multiple metrics and scenarios. This effectiveness translates successfully to production environments, as demonstrated through online A/B tests, and has been deployed in a commercial system supporting hundreds of millions of daily users.

To sum up, our main contributions include the design and implementation of a unified, user-centric interest modeling framework that achieves both expressiveness and efficiency for billion-scale cross-scenario recommendation, complemented by a comprehensive million-scale multi-scenario sequential dataset that enables rigorous evaluation of unified modeling approaches. Through empirical validation in both offline and online production environments, we demonstrate substantial improvements that establish the practical viability of unified cross-scenario modeling at billion scale.

## 2 RELATED WORK

**Sequence Modeling** The advent of deep learning has greatly advanced recommender systems, with methods such as neural collaborative filtering (He et al., 2017) and factorization machines (Rendle, 2010; Guo et al., 2017) excelling at capturing intricate user–item interactions. Sequential recommendation systems have evolved from simple recurrent architectures to sophisticated transformer-based models capable of learning complex temporal dependencies in user behavior. Early work such as GRU4Rec (Hidasi et al., 2015) pioneered the use of recurrent neural networks for modeling session-based interactions. Subsequent models like Caser (Tang & Wang, 2018) employed convolutional filters to extract both short- and long-term patterns, while attention-based methods further improved the ability to focus on relevant historical interactions. Transformer-based approaches, including SASRec (Kang & McAuley, 2018) and BERT4Rec (Sun et al., 2019), introduced self-attention mechanisms to capture long-range dependencies, and bidirectional encodings to leverage full context for superior representation learning. More recent advances have explored contrastive learning (Zhou et al., 2020; Wei et al., 2023), multi-interest modeling (Li et al., 2019; Cen et al., 2020), and graph neural networks (Wang et al., 2020; Zhang et al., 2022b; Yang et al., 2023) to better model the dynamic and multifaceted nature of user preferences. Emerging research has begun leveraging large language models to enhance user and item representations in sequential recommendation tasks (Chen et al., 2024a; Hu et al., 2024; Wang et al., 2024b), as well as generative patterns (Chen et al., 2024b; Paischer et al., 2024; Deng et al., 2025; Han et al., 2025) that bridge natural language understanding and recommendation tasks. This paradigm shift has brought about transformative improvements in recommendation pipelines.

**Multi-scenario Recommendation** While user interests are generally stable, behavioral patterns can vary significantly across scenarios (Zang et al., 2022; Gao et al., 2023). Early cross-platform studies (Niu et al., 2021; Tan et al., 2021) found that users maintain similar topical interests on different platforms, despite differences in interaction patterns and frequencies. These insights inspired methods to disentangle latent interests from observed behaviors. Multi-scenario recommendation methods (Tan et al., 2021; Zhao et al., 2023; Li et al., 2024; Wu et al., 2025) model diverse user behaviors while capturing shared interest representations. For instance, M2M(Zhang et al., 2022c) introduced novel meta units for ads scenarios. Advances in disentangled representation learning have better separated stable interests from contextual actions; works like STAR(Sheng et al., 2021), AdaSparse(Yang et al., 2022) and APG(Yan et al., 2022) employ domain-aware designs to learn domain commonalities and distinctions in CTR prediction. HierREC(Gao et al., 2024) combines explicit and implicit scenario-aware modules to capture hybrid-grained information. Graph-based methods (Tan et al., 2021; Cao et al., 2022) are widely used to model multi-behavioral patterns and generate unified user embeddings. This research direction is also known as cross-domain (Ma et al., 2022) or multi-domain (Zhao et al., 2023; Yang et al., 2024) recommendation, leveraging multi-source user histories to enhance performance. However, scalability remains an issue for most approaches on industrial-scale datasets, and large-scale online validations are limited. Recently, methods utilizing foundation models to build universal recommenders have been proposed (Wang et al., 2024a; Shen et al., 2024), including works for multi-scenario settings like LLM4CDSR(Liu et al., 2025), but they have yet to be extended to billion-scale industrial settings.

## 3 CROSS-SCENARIO DATASET

Existing open-source sequential recommendation datasets are constrained by their narrow focus, often capturing user behaviors in isolated e-commerce or entertainment scenarios centered around singular interaction types such as ratings, clicks, or purchases (Ben-Shimon et al., 2015; Harper & Konstan, 2015; Ni et al., 2019; Zhu et al., 2018). These datasets, while foundational for earlier research, are fundamentally limited in their ability to model the multi-scenario, and cross-modal nature of user interests observed in modern large-scale UGC platforms. Notably, even in most recent datasets such as KuaiRand (Gao et al., 2022) and Qilin (Chen et al., 2025) that begin to characterize UGC environments, a fragmented approach is frequently adopted that underrepresents the complex interplay between scenarios and only partially reflects the holistic evolution of user interests.

To address these limitations, we first introduce a multi-scenario sequential recommendation dataset derived from billions of user interactions from a third-party UGC platform, featuring several key characteristics:

**Diverse Behavioral Contexts** Our dataset encompasses a comprehensive range of real-world interaction scenarios on the UGC platform, including (a) homefeed browsing, (b) search-driven browsing and clicking, and (c) ad exposure and engagement. This temporally-aligned diversity enables robust analysis of user behavior across distinct yet interconnected scenarios within a unified platform ecosystem.

**Comprehensive Engagement Patterns** The dataset uniquely captures both explicit positive user engagements such as clicks, likes, collections, and shares, as well as negative signals. Additionally, we record view duration for each interaction, providing a nuanced and holistic depiction of user preferences and attention patterns.

**Industrial Coverage** The dataset includes millions of items and hundreds of millions of engagement records, surpassing existing datasets in scale and delivering the volume and complexity needed for developing and benchmarking cutting-edge recommender models at scale. Also, by tracking user behavior over extended time periods, our dataset facilitates the study of long-term interest evolution, behavioral stability, and cross-scenario consistency, which are typically constrained in other publicly available datasets.

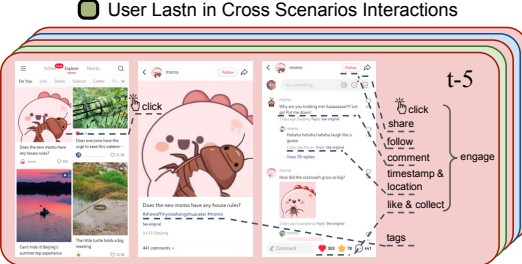

(a) Scenario 1: Homefeed at time t-5

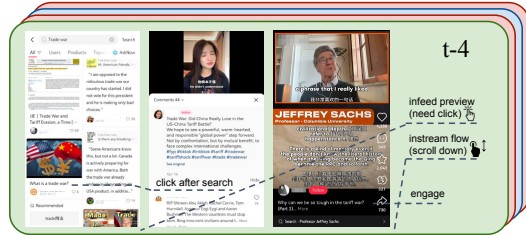

(b) Scenario 2: Search at time t-4

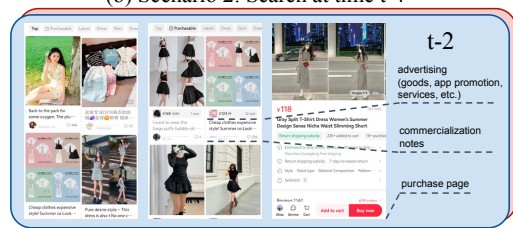

(c) Scenario 3: Advertisements at time t-2

Figure 2: **Multiple scenarios represented in the dataset.** Our work primarily focuses on three key scenarios: homefeed, search, and ads.

An example datapoint is shown in Figure 2, while Figure 3 presents overall dataset statistics. Additional details including dataset collection and filtering can be found in Section C.

## 4 MULTI-SCENARIO USER INTEREST LEARNING

### 4.1 TASK FORMULATION

Let $\mathcal{U} = \{u_1, u_2, \ldots, u_N\}$ denotes the set of users and $\mathcal{I} = \{i_1, i_2, \ldots, i_M\}$ represents the universal item space. The item space contains posts with image-text or video content created by either regular users or advertisers, which can be recommended to users or discovered through search interactions.

For each user $u \in \mathcal{U}$, we observe a chronologically ordered engagement sequence $S_u = \{(i_1, a_1, s_1, t_1), (i_2, a_2, s_2, t_2), \ldots, (i_{|S_u|}, a_{|S_u|}, s_{|S_u|}, t_{|S_u|})\}$, where each interaction tuple consists of $i_t \in \mathcal{I}$ (the interacted item at timestamp $t$), $a_t \in \mathcal{A}$ (the engagement action performed), $s_t \in \mathcal{S}$ (the scenario context), and $t$ (the interaction timestamp). The engagement sequence encompasses user activities across three distinct scenarios $\mathcal{S} = \{\text{homefeed, ads, search}\}$. [1] Each interaction $i_t$ is associated with rich contextual information including content features, and user engagement actions $a_t \in \mathcal{A}$, where $\mathcal{A} = \{\text{like, share, comment, follow, messaging, block}\}$ represents the comprehensive set of possible user responses to content.

---

[1]While advertisements can appear in search results for specific keywords, we consider such interactions as highly target-specific and exclude them from this task scope.

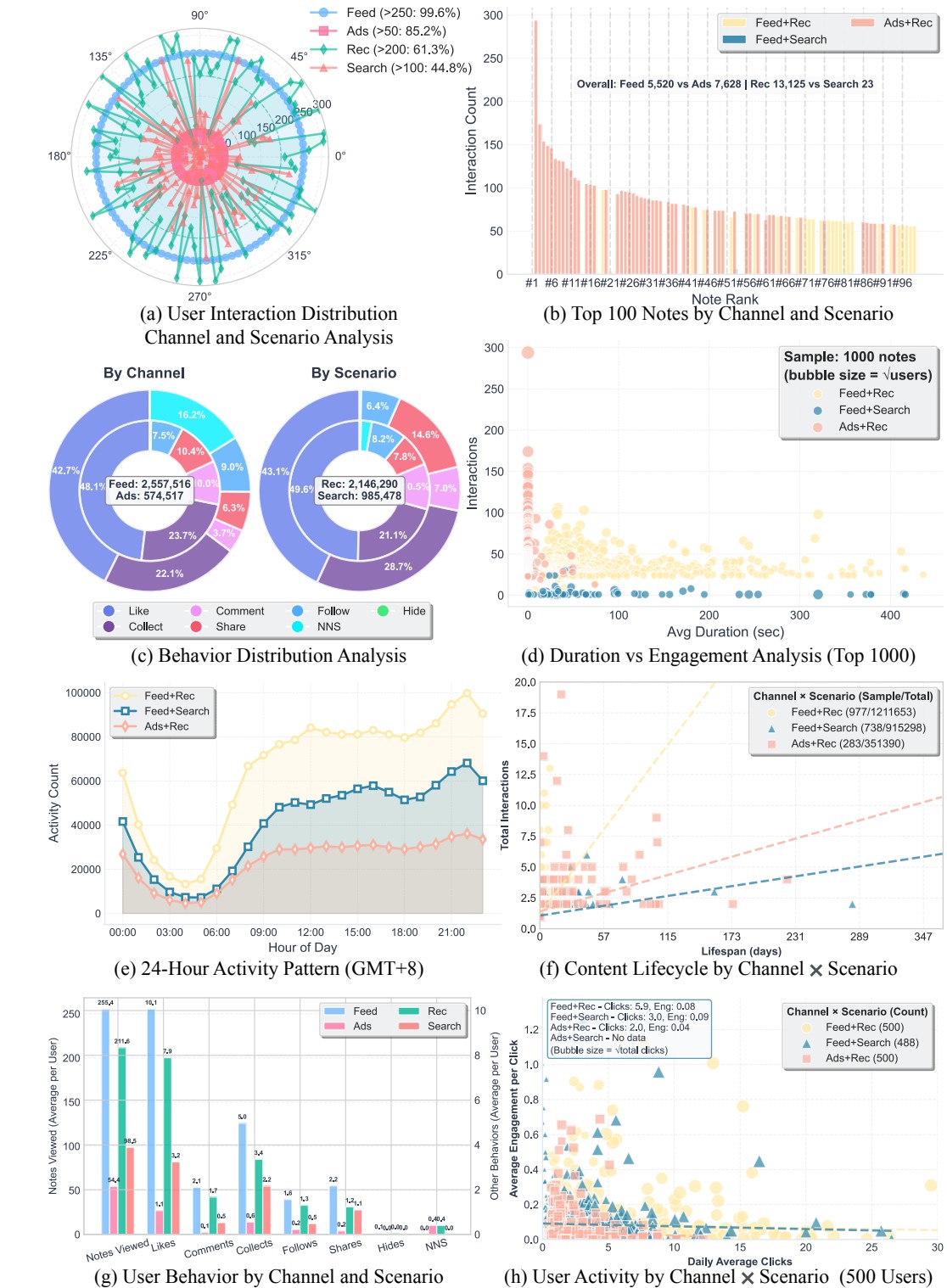

Figure 3: **User engagement analytics dashboard.** We sample 10k users for a comprehensive visualization of their interaction patterns across feed & ads channels and recommendation & search scenarios, focusing on their behavioral trends, content performance, and engagement metrics: (1) User distribution by channel & scenario; (2) Top notes engagement analysis; (3) Behavior pattern distribution; (4) Duration vs. engagement correlation; (5) 24-hour activity trends; (6) UGC content lifecycle included; (7) User behavior by channel & scenario; (8) User activity by channel & scenario.

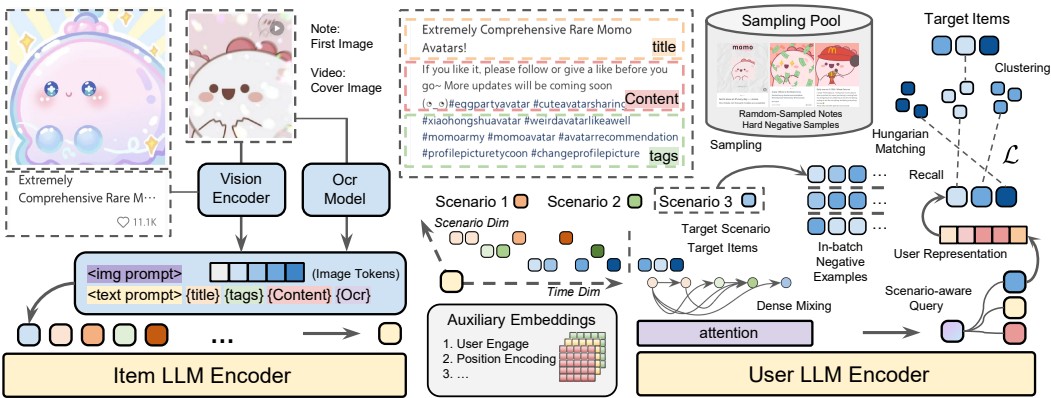

Figure 4: **Overall framework of *RED-Rec*.** *RED-Rec* is a two-tower hierarchical architecture comprising a multi-modal item encoder and a sequential user encoder. The item encoder fuses textual and visual signals into unified embeddings, while the user encoder utilizes scenario-specific transformer blocks to model the evolution of user interests. The system is trained end-to-end.

Given the multi-scenario user engagement sequences, our primary objective is to learn unified representations that capture user preferences and item characteristics across different contexts. Formally, we aim to learn embedding functions $f_u : \mathcal{U} \times \mathcal{H}_u \rightarrow \mathbb{R}^d$ and $f_i : \mathcal{I} \rightarrow \mathbb{R}^d$ that map users (conditioned on their interaction history) and items to a shared $d$-dimensional embedding space:

$$\mathbf{u} = f_u(u, S_u), \quad \mathbf{v}_i = f_i(i)$$

where $\mathbf{u} \in \mathbb{R}^d$ represents the user embedding and $\mathbf{v}_i \in \mathbb{R}^d$ represents the item embedding. The embedding can be directly used in Recall Task or used as features for downstream models like pre-ranking and fine-grained ranking.

## 4.2 HIERARCHICAL LLM-BASED REPRESENTATION LEARNING

*RED-Rec* is a two-tower LLM-powered framework designed to learn comprehensive item and user embeddings. It incorporates a 2-D dense mixing policy for effectively aggregating and fusing user interactions across diverse scenarios, along with multi-scenario multi-interest querying to capture various facets of user preferences. The framework operates by first encoding items through a dedicated item encoder, then fusing scenario-specific interactions which are subsequently processed by the user encoder to generate tailored user embeddings for each scenario.

**Item Representation Learning** For each item $i \in \mathcal{I}$, we employ a multi-modal encoder $\mathrm{E}_{\text{item}}$ to generate an embedding $\mathbf{e}_i = \mathrm{E}_{\text{item}}(\mathbf{x}_i, \mathbf{v}_i; \theta_i, \theta_v) \in \mathbb{R}^d$, where $\mathbf{x}_i$ is the item's textual input—consisting of title, tags, content, and OCR-extracted text—and $\mathbf{v}_i$ is its visual content. Textual features are encoded by a pre-trained language model (parameters $\theta_i$), while images are processed by the ViT (Dosovitskiy et al., 2020) vision encoder (parameters $\theta_v$) and then projected to dimension $d$ via a linear layer.

**Sequential User Interest Modeling** At timestamp $t$, we model the user's current interest based on their past $n$ behaviors across the platform. Given user $u$'s combined interaction sequence $S_u = S_u^h \cup S_u^a \cup S_u^s$, where $S_u^h$ denotes homefeed interactions, $S_u^a$ denotes advertising interactions, and $S_u^s$ denotes search interactions. Each interaction is represented as $S = \{\text{Content, timestamp, action}\}$. The current sequence representation can be denoted as $\mathbf{H}_u = [\mathbf{e}_{i_1}, \mathbf{e}_{i_2}, \ldots, \mathbf{e}_{i_n}] \in \mathbb{R}^{n \times d}$ where $\mathbf{e}_{i_t}$ represents the content embedding for the $t$-th interaction in the user's recent history.

Based on this, actions $\mathbf{A}_u = [\mathbf{a}_{i_1}, \mathbf{a}_{i_2}, \ldots, \mathbf{a}_{i_n}]$ are represented as one-hot vectors encoding engagement behaviors including {collect, share, message, block, like} for interaction $n$. These action vectors are converted to dense embeddings using a learnable embedding layer. Furthermore, timestamps are not only used to sort engagement chronologically but also serve as temporal features at the hour level, as we observe that user interests and behaviors vary significantly across different times of the day. Specifically, each timestamp is discretized into a 24-dimensional one-hot vector representing the hour of the day $\mathbf{h}_{i_t} = \text{OneHot}(\text{hour}(timestamp_t)) \in \{0, 1\}^{24}$, and then converted to

dense embeddings through a learnable embedding layer. The final enhanced representation $\hat{\mathbf{H}}_u$ incorporates content, action, and temporal information. The user-level encoder processes this enriched sequence to generate contextualized user representations $\mathbf{U}_u = \mathrm{E}_{\mathrm{user}}(\hat{\mathbf{H}}_u; \theta_u)$.

**Cross-scenario Interest Mix and Query**    To effectively capture user interests across multiple scenarios, we introduce a cross-scenario 2-D dense mixing and querying policy in *RED-Rec*.

The mixing policy serves as a gating and fusion mechanism, aggregating user behaviors from multiple scenarios (homefeed, ads, search) before the user encoder. Formally, let $S_u^s[-n_s :]$ denote user $u$'s latest $n_s$ records in scenario $s \in \{\text{homefeed}, \text{ads}, \text{search}\}$. We define the mixed sequence:

$$S_u^{\mathrm{mixer}} = \mathrm{Merge}\Big( S_u^{\mathrm{homefeed}}[-n_h :], \ S_u^{\mathrm{ads}}[-n_a :], \ S_u^{\mathrm{search}}[-n_s :] \Big), \tag{1}$$

where $\mathrm{Merge}(\cdot)$ deterministically fuses events—sorting by timestamp and concatenating per a fixed scenario order, retaining scenario tags. During training, $n_s$ samples per scenario are used; at inference, the most recent $n_s$ are selected, with $n_s$ tuned as a hyperparameter. "2-D dense mixing" refers to filtering events along both scenario (balancing quotas) and temporal (recency) axes, preserving all selected events for encoding. This addresses behavior imbalance (e.g., far more homefeed than ad/search actions), ensuring that infrequent but informative user signals are retained for downstream modeling. We have ablations for different mixing policies in Section 5.

To further enhance modeling capacity, we design a 2-D positional encoding for each event. For the $j$-th event in $S_u^{\mathrm{mixer}}$, we define 1) Sequence position encoding as $\mathbf{PE}_{\mathrm{seq}}(j)$, reflecting the event's absolute position in the sequence; and 2) Time-gap encoding as $\mathbf{PE}_{\mathrm{gap}}(\Delta t_j)$, where $\Delta t_j = t_{\mathrm{curr}} - t_j$ is the interval between the interaction and the present. The final positional encoding is computed as $\mathbf{p}_j = \mathbf{PE}_{\mathrm{seq}}(j) + \mathbf{PE}_{\mathrm{gap}}(\Delta t_j)$.

The query module enables learnable query embeddings, termed *scenario-aware queries*, denoted as $\mathbf{Q} = [\mathbf{q}_1, \mathbf{q}_2, \ldots, \mathbf{q}_K] \in \mathbb{R}^{K \times d}$, where $K$ is the number of interest aspects. These queries enable the model to attend to different facets of user preference under varying contexts. The scenario-aware user representation is then formed by feeding the concatenation of user interaction history (excluding the most recent $W$ actions) and $\mathbf{Q}$ into a user encoder:

$$\mathbf{U}_u^{\mathrm{query}} = \mathrm{E}_{\mathrm{user}}\left( \left[ \tilde{\mathbf{H}}_u[: -W]; \ \mathbf{Q} \right]; \theta_u \right), \tag{2}$$

where $W$ denotes the window size for recent interactions. By leveraging different queries, the user encoder can generate multiple representations, each reflecting a distinct aspect of user interest.

During training, we adopt *Noise Contrastive Estimation* (NCE) as the main objective to model sequential recommendation. Given refined interest embeddings $\{\mathbf{r}_1, \ldots, \mathbf{r}_s\}$ and positive samples $\{\mathbf{t}_1, \ldots, \mathbf{t}_w\}$ from the target window, we cluster the positive samples into $s$ groups using cosine similarity and then match cluster centroids to interest embeddings via the Hungarian algorithm(Kuhn, 1955) to maximize pairwise similarity. The contrastive loss is applied only to these matched pairs:

$$\mathcal{L}_{\mathrm{total}} = \frac{1}{w} \sum_{i=1}^{w} \sum_{j=1}^{s} \mathcal{L}_{\mathrm{NCE}}(t_i, r_j) \cdot \Pi(i, j) \tag{3}$$

where $\Pi(i, j) = 1$ if the cluster of $t_i$ is matched with $r_j$, and 0 otherwise.

### 4.3 BENCHMARKING AND OFFLINE EVALUATION

We evaluate representations on recall tasks with a temporal split protocol. For each user $u$, their interaction sequence $S_u$ is split at a randomly sampled cutoff $t_{\mathrm{cut}}$. The input is $S_u^{\mathrm{input}} = \{(i, a, s, t) \in S_u : t < t_{\mathrm{cut}}\}$, and the next three items form the targets: $\mathcal{G}_u = \{i_{t_{\mathrm{cut}}}, i_{t_{\mathrm{cut}}+1}, i_{t_{\mathrm{cut}}+2}\}$. A candidate pool $\mathcal{C}$ is formed by random sampling from the active items in the platform and mixing in $\mathcal{G}_u$. The user embedding $\mathbf{u}$ is computed from $S_u^{\mathrm{input}}$, and similarity scores are defined as $\mathrm{score}(u, i) = \cos(\mathbf{u}, \mathbf{v}_i), \quad \forall i \in \mathcal{C} \cup \mathcal{G}_u$. Items are ranked to produce recommendations $\mathcal{R}_u$, and top-$K$ results $\mathcal{R}_u^K$ are evaluated using HR@K, NDCG@K, and MRR for $K \in \{10, 50, 100, 1000\}$.

Table 1: **Recommendation results for individual scenarios.** Baseline methods are evaluated separately on the homefeed and ads scenarios. In all metrics, higher scores reflect better performance.

| | Homefeed | | | | Ads | | | |
|---|---|---|---|---|---|---|---|---|
| Baselines ↑ | HR/NDCG$_{10}$ | HR/NDCG$_{100}$ | HR/NDCG$_{1k}$ | MRR$_{*100}$ | HR/NDCG$_{10}$ | HR/NDCG$_{100}$ | HR/NDCG$_{1k}$ | MRR$_{*100}$ |
| SASRec | 1.76/0.97 | 12.32/1.79 | 32.01/4.04 | 1.01 | 3.26/1.63 | 14.08/3.71 | 39.11/5.27 | 1.57 |
| MoRec | 1.78/**1.25** | 12.48/**2.23** | 31.98/**4.12** | 1.21 | 3.47/1.67 | 13.98/**3.88** | 38.27/4.89 | **1.78** |
| HSTU | **1.79**/1.22 | 12.72/2.21 | 31.76/3.69 | 1.15 | 3.85/1.70 | 14.32/3.30 | 38.20/**5.38** | 1.43 |
| HLLM | 1.66/0.62 | **12.77**/1.83 | **32.52**/4.02 | **1.22** | **4.21**/1.21 | 14.27/3.37 | **39.21**/4.48 | 1.39 |
| DLRM-v3 | 1.63/1.03 | 11.33/2.01 | 28.96/3.72 | 1.13 | 3.54/1.21 | **15.27**/3.22 | 35.39/4.27 | 1.67 |
| *RED-Rec* | 2.31/0.68 | 12.59/1.88 | 31.94/3.86 | 1.27 | 4.24/1.28 | 16.44/3.21 | 40.18/4.61 | 1.96 |
| *RED-Rec*-pt | 2.90/0.63 | 14.89/2.02 | 36.16/4.01 | 1.30 | **4.84**/**1.30** | 17.66/2.87 | **42.71**/**5.21** | **2.27** |
| *RED-Rec*-mm | 2.35/1.21 | 14.20/**2.27** | 31.29/3.97 | 1.29 | 4.31/1.31 | 17.22/3.18 | 41.86/4.66 | 1.92 |
| *RED-Rec*-mm-pt | **3.23**/**1.27** | **15.46**/2.21 | **36.29**/**4.14** | **1.38** | 4.82/1.19 | **18.21**/3.29 | 42.56/4.98 | 2.21 |

# 5 EXPERIMENTS

## 5.1 EXPERIMENTAL SETTINGS AND BASELINES

In our default setup, all models are trained on a standardized dataset of 1 million users and evaluated on 10,000 test samples, using a randomly sampled base pool of approximately 1 million notes for fair comparison. We set the window size W=10, last_n=128, and use 3 queries per scenario. Both item and user encoders are initialized with either a 1.3B-parameter LLaMA-based model (Cui et al., 2023) or the 1.5B Qwen-2.5 model (Yang et al., 2025), while the vision encoder uses CLIP ViT-B/16. Training requires about 24 hours on 8 NVIDIA H100 GPUs for 3 epochs, with batch size 2 and gradient accumulation of 4. Detailed implementation can be found in Section E.

We compare our model with several mainstream baselines: SASRec (Kang & McAuley, 2018), MoRec (Yuan et al., 2023), HSTU (Zhai et al., 2024), HLLM (Chen et al., 2024a), and DLRM-v3 (Naumov et al., 2019). Experiments are conducted in single-scenario (homefeed, ads) and multi-scenario (e.g., search + homefeed, homefeed + ads, all combined) settings. Our model is evaluated in four variants: *RED-Rec*-symbol, *RED-Rec*-mm, and their pre-trained versions ("-pt"), the latter trained on a large-scale online dataset. Standard recommendation metrics are used (see Section D). We evaluate all baselines on our proposed industrial dataset to demonstrate the performance gains brought by unified modeling. Meanwhile, experiments on a public dataset are presented in Section B.

Table 2: **Recommendation results for mixed scenarios.**

| **Search + Homefeed** (for Homefeed) | | | |
|---|---|---|---|
| Baselines ↑ | HR/NDCG$_{10}$ | HR/NDCG$_{100}$ | HR/NDCG$_{1k}$ | MRR$_{*100}$ |
| SASRec | 1.73/1.22 | 12.02/3.21 | 32.17/4.17 | 1.52 |
| MoRec | 1.79/1.30 | 13.92/2.99 | 33.01/3.98 | 1.53 |
| HSTU | 1.79/1.25 | 12.84/3.28 | 33.15/4.24 | 1.55 |
| HLLM | 1.69/1.02 | 13.49/3.18 | 33.04/4.21 | 1.58 |
| DLRM-v3 | 1.64/1.18 | 11.35/3.02 | 30.89/3.98 | 1.48 |
| *RED-Rec* | 2.26/1.32 | 14.74/3.16 | 33.29/4.20 | 1.58 |
| *RED-Rec*-pt | **2.92/1.33** | **18.26/3.24** | **38.92/4.23** | **1.67** |
| **Homefeed + Ads** (for Ads) | | | |
| Baselines ↑ | HR/NDCG$_{10}$ | HR/NDCG$_{100}$ | HR/NDCG$_{1k}$ | MRR$_{*100}$ |
| SASRec | 3.72/1.24 | 16.18/3.08 | 38.94/4.72 | 1.94 |
| MoRec | 3.80/1.30 | 17.23/2.62 | 38.29/4.77 | 1.98 |
| HSTU | 3.89/1.28 | 16.95/3.15 | 40.12/4.81 | 2.01 |
| HLLM | 3.68/1.19 | 17.24/3.12 | 39.76/4.78 | 1.97 |
| DLRM-v3 | 3.52/1.21 | 15.43/2.95 | 36.87/4.58 | 1.87 |
| *RED-Rec* | 4.36/1.31 | 18.32/3.27 | 42.61/5.02 | 2.11 |
| *RED-Rec*-pt | **5.18/1.38** | **18.89/3.21** | **46.59/5.57** | **2.38** |
| **Homefeed + Search + Ads** (for Ads) | | | |
| Baselines ↑ | HR/NDCG$_{10}$ | HR/NDCG$_{100}$ | HR/NDCG$_{1k}$ | MRR$_{*100}$ |
| SASRec | 3.68/1.21 | 14.29/2.08 | 38.94/4.72 | 1.94 |
| MoRec | 3.82/1.33 | 18.27/2.98 | 38.41/4.66 | 1.98 |
| HSTU | 3.92/1.31 | 17.21/3.19 | 40.14/4.81 | 2.11 |
| HLLM | 4.08/1.11 | 19.92/3.18 | 43.27/4.91 | 2.06 |
| DLRM-v3 | 3.34/1.01 | 14.08/2.81 | 35.74/4.36 | 1.74 |
| *RED-Rec* | 4.72/1.33 | 18.33/3.22 | 42.89/4.97 | 1.94 |
| *RED-Rec*-pt | **5.18/1.35** | **20.52/3.24** | **49.17/5.93** | **2.41** |

## 5.2 SINGLE-SCENARIO

We first evaluate our model alongside baseline methods on two distinct recommendation scenarios: the Homefeed and Ads settings, as presented in Table 1. The results demonstrate that *RED-Rec* outperforms popular baselines, even in scenarios that were not specifically targeted during optimization. We attribute this improvement both to the modeling of multi-interest user representations and to advancements in the backbone framework. Compared to baselines relying on ID-based representations

like SASRec and HSTU, leveraging LLMs provides richer semantic encodings, especially beneficial under cold-start conditions, resulting in substantial performance gains. Furthermore, when compared to models with similar architectures, such as HLLM, our approach benefits from a larger backbone with enhanced Chinese language capabilities, allowing for a better fit to the dataset and further boosting effectiveness.

### 5.3 MULTI-SCENARIO

We further evaluate our model in multi-scenario settings to assess whether incorporating information flow between scenarios can enhance performance, particularly in the homefeed and ads scenarios (Table 2). Additionally, we illustrate the improvements brought by multi-scenario recommendation in Figure 5. The most significant gains are observed in two specific cases: leveraging data from search scenario improves homefeed recommendation, while utilizing both homefeed and search data enhances ads recommendation.

We observe that modeling information flow between scenarios—such as integrating Homefeed, Ads, and Search data—consistently improves the performance of most baselines, with the largest gains seen for *RED-Rec*. This can be attributed to its effective multi-source signal integration and the advanced user-side LLM, which offers strong few-shot reasoning capabilities. For Homefeed recommendations, access to Search data—especially post-search clicks—significantly increases HR and NDCG scores. Similarly, Ads recommendations benefit from combined Homefeed and Search behaviors, showing the greatest metric improvements. These enhancements are consistent across all cutoff values, and *RED-Rec* not only raises the likelihood of relevant items being recommended, but also ensures they appear closer to the top.

### 5.4 MODULE ABLATION

We further conduct ablation studies on several key components of the model design, focusing on the basic model capabilities: (1) input sequence length, (2) multi-interest query, and (3) pretraining on a larger-scale dataset. For cross-scenario recommendation, we also examine different methods for the mixer module to combine various input sources. The ablation

Table 3: **Model ablation for *RED-Rec***. Top: model config ablation; Bottom: scenario mixing policy ablation.

| Homefeed | | | |
|---|---|---|---|
| Setting | HR/NDCG$_{10}$ | HR/NDCG$_{100}$ | HR/NDCG$_{1k}$ | MRR |
| SeqLen = 128, Multi-Interest, pt | **2.90**/0.63 | **14.89/2.02** | **36.16**/4.01 | 1.30 |
| SeqLen = 128, Multi-Interest | 2.31/0.68 | 12.59/1.88 | 31.94/3.86 | 1.27 |
| SeqLen = 128, Single-Interest | 1.85/**0.72** | 10.24/1.95 | 26.78/**4.12** | **1.31** |
| SeqLen = 64, Multi-Interest | 2.08/0.71 | 11.32/1.94 | 28.67/3.92 | 1.29 |
| SeqLen = 32, Multi-Interest | 1.72/0.61 | 9.48/1.76 | 25.47/3.64 | 1.29 |
| Homefeed + Search + Ads | | | |
| Mixer Strategy | HR/NDCG$_{10}$ | HR/NDCG$_{100}$ | HR/NDCG$_{1k}$ | MRR |
| Sorted by Timestamp | 2.10/0.53 | 10.55/1.90 | 21.44/2.20 | 0.65 |
| Naive Combination | 4.28/1.22 | 17.60/3.06 | 41.90/4.85 | 1.85 |
| 1D (on position) | 4.31/1.23 | 17.65/3.08 | 41.95/4.87 | 1.86 |
| 1D (on timestamp) | 4.40/1.25 | 17.80/3.10 | 42.20/4.90 | 1.88 |
| 2D-Mixing (*RED-Rec*) | **4.72/1.33** | **18.33/3.22** | **42.89/4.97** | **1.94** |

results (Table 3) show that longer input sequences, multi-interest queries, and large-scale pretraining all lead to improved recommendation metrics. For cross-scenario settings, our 2D-mixing policy yields the strongest performance, highlighting the value of integrating positional and temporal information for effective signal fusion.

### 5.5 MODEL SCALING

The choice of the 1.5B Qwen2.5 model for initialization balances model accuracy with online deployment costs. To investigate the effect of model size, we train multiple models—each on the same number of tokens—and evaluate them on an offline test set, exploring scaling laws relative to parameter count. We examine two model families: LLaMA (Touvron et al., 2023) (mainly LLaMA 2 series) and Qwen (Yang et al., 2025) (mainly Qwen 2.5 series), with model sizes ranging from 0.5B to 7B parameters. As shown in Figure 5 (b), we report both Hit Rate (HR) performance and the corresponding Sample per Second (SPS) within our deployment environment for the Homefeed+Search+Ads scenario. Results show that larger model sizes consistently improve HR up to 7B parameters in both families, indicating the potential for scaling law benefits. However, increased model size leads to decreased SPS, limiting the feasibility of deploying larger models for online serving. Throughput results, as measured by SPS, are also presented in Figure 5 (b).

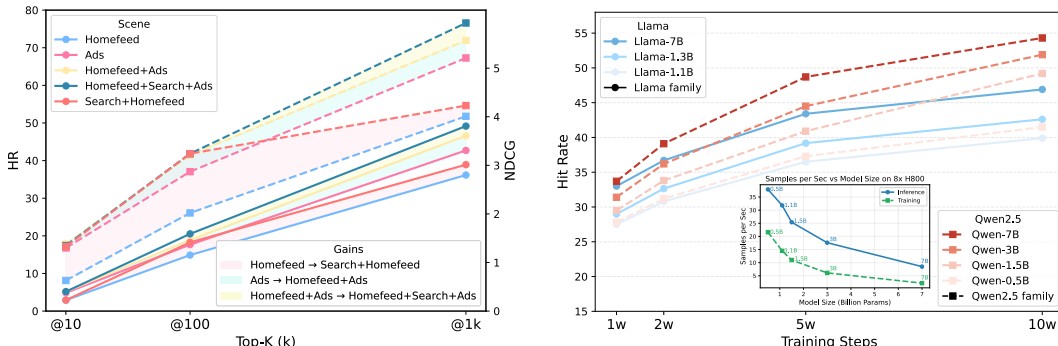

Figure 5: **Comparison of scaling laws and gains from multi-scenario inclusion.** (a) Performance gains achieved through unified modeling with *RED-Rec*, where the shaded region represents the improvement; (b) Scaling behavior of model size with respect to click HR.

## 5.6 ONLINE SERVING

We conducted online A/B experiments on downstream recommendation scenarios within the recall stage of an industrial recommender system. We specifically report results in advertising scenarios, as offline studies demonstrate the most significant improvements when generalizing user interests from homefeed click patterns. The experiment employed a balanced traffic allocation of 10% treatment versus 10% control, running for one week. The item pool consists of active items across the platform (approximately 1.1 billion) and was tested against the full active user base (approximately 160 million users).

*RED-Rec* achieved a 0.8864% improvement in total *ADVV* and a 0.3401% increase in overall Feed Ad Spend (*Cost*), representing substantial improvements given the platform's scale. Notably, over 90% of items recalled by our approach are unique within the initial candidate generation phase, demonstrating significant incremental value and diversity to the recommendation pool.

Table 4: **A/B Test Results Comparing Experimental vs Control Groups.** We report key metrics in different domains.

| Domain | Metric | Change (%) |
|---|---|---|
| Community | APP_LT (long-term active users) | -0.0015 |
| | SAU_LT (search-related) | -0.0007 |
| E-commerce | Overall Purchase UV | -0.0886 |
| Live Streaming | Live Broadcast Engagement UV | -0.0694 |
| Commercialization | Advertiser Value (*ADVV*) | +0.8864 |
| | Advertising Spend | +0.3401 |

We observe minimal trade-offs: platform-level content engagement declined marginally (less than 0.01% in most cases), with small reductions in purchase UV (-0.09%) and live broadcast engagement (-0.07%) due to systematic ad prioritization based on learned user preferences. These negligible decreases are substantially outweighed by advertising performance gains while maintaining user experience integrity. Based on these promising results, we have fully deployed the method to production across the entire platform.

Details of our online serving can be found in Section E.2.

## 6 CONCLUSION

We introduce *RED-Rec*, a unified hierarchical LLM-based sequential recommendation framework designed to leverage multi-scenario behavioral contexts for context-aware user modeling at industry scale. The proposed two-tower architecture, combined with scenario-aware mixing and querying policies, enables expressive and efficient recommendations across diverse scenarios, including feeds, search, and advertising. Comprehensive empirical evaluations—conducted on a newly constructed million-scale multi-scenario dataset and through large-scale real-world deployment—demonstrate substantial improvements over strong baselines in both offline and production environments. Our work underscores the importance of unified user interest modeling in enabling more consistent, intelligent, and user-centric recommendation systems, paving the way for richer and more seamless experiences on large-scale content platforms.

ETHICS STATEMENT

The dataset used and planned for release in this work has been fully anonymized and does not contain any personal or individually identifiable information, but rather consists of a collection of publicly accessible content. The paper does not include any analysis, reporting, or disclosure of private user details, and care has been taken to ensure that all data handling aligns with privacy regulations and ethical guidelines.

REPRODUCIBILITY STATEMENT

We have provided demo source code and running tutorials as supplementary materials and also in https://anonymous.4open.science/r/RedSeqRec-ano-4158. Key implementation details and experimental settings are described in the main paper (Section 4 and Section 5). We promise we will open-source both the code and the dataset used in our experiments upon acceptance.

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

## A    TERMS

We would like to firstly offer additional explanations for specific terms used throughout the paper in order to facilitate understanding for non-expert readers:

**Homefeed**    refers to the main feed or landing page displayed to a user when they open a content platform or app. It typically consists of a personalized selection of items (such as posts, products, videos, etc.) recommended to the user based on their preferences and past behavior.

**Internal Flow**    denotes the content consumption pattern within the single-column sliding or swiping through content (e.g., images, videos, or articles). Users engage with recommendations directly within this detailed view by navigating between related items or sliding to the next recommended content.

**External Flow**    refers to the content consumption flow that occurs on the main feed of the platform, where users browse the list of recommended items presented to them upon opening the app. This process typically involves users scrolling vertically through the two-columns page.

**Scenarios**    refer to distinct user interaction environments or channels within the platform, each characterized by unique user intents and behavioral patterns. In this paper, we focus on three core scenarios: **homefeed**, **ads**, and **search**. The homefeed scenario represents the primary personalized feed where users consume a diverse assortment of recommended content. The ads scenario corresponds to user engagement with sponsored or promotional content distributed throughout various parts of the platform. Although advertisement content can appear within the homefeed, we treat it as a separate scenario because it represents a different source and serves distinct business objectives. The search scenario involves users actively retrieving information or content by submitting queries.

**last-$n$**    refers to the most recent 'n' items a user has interacted with on the platform. For example, 'last10' indicates the user's last 10 consumed items. This concept is commonly used to capture and analyze a user's most current interests or activity history.

**Engage**    represents user interactions with content, such as clicks, likes, comments, shares, or dwell time. Engagement metrics are used to measure how users interact with recommended items and to assess the effectiveness of recommender systems.

## B    FURTHER EXPERIMENTS

We further test *RED-Rec* on Amazon Books Reviews(McAuley et al., 2015), a widely used subset in recommender system research datasets which sampled from the Amazon Review dataset. In the Books subset, each review typically contains fields such as reviewer ID, item (book) ID, rating (1-5 stars), review text, timestamp, and sometimes additional metadata (e.g., book title). We test and compare *RED-Rec* in Table 5. Following LLM4CDSR(Liu et al., 2025), we further test and compare *RED-Rec* in an cross-scenario setting, where we choose the Cloth-Sport sub-categories of the Amazon dataset. The results are shown in Table 6 [2].

On the Amazon Books Reviews dataset, our model achieves slightly better results than HLLM and outperforms traditional methods. A reasonable explanation is that both our approach and HLLM leverage LLMs for semantic-based item embeddings, which provides a natural advantage in capturing deeper item relationships. In the cross-scenario reasoning task, *RED-Rec* also surpasses most baselines. Although it performs slightly worse than LLM4CDSR in some cases, it is important to note that *RED-Rec* and LLM4CDSR are fundamentally designed for different task settings. Specifically, LLM4CDSR focuses more on interest generalization across distinct interest groups (such as from Clothing to Sports) through careful prompt and architectural design, while *RED-Rec* aims for large-scale unified modeling across business domains, focusing more on the holistic user modeling from their behaviors.

---

[2]* means results of the baselines are adopted from the original LLM4CDSR paper.

Table 5: A comparison on the Amazon Books dataset.

| Baselines ↑ | HR/NDCG$_{10}$ | HR/NDCG$_{50}$ | HR/NDCG$_{200}$ |
|---|---|---|---|
| SASRec | 3.06/1.64 | 7.54/2.60 | 14.31/3.62 |
| MoRec(bert) | 3.21/1.82 | 8.21/2.33 | 18.29/3.71 |
| HSTU-1B | 4.78/2.62 | 10.82/3.93 | 19.08/5.17 |
| DLRM-v3-1B | 6.22/2.88 | 12.74/5.12 | 23.12/5.29 |
| HLLM-1B | 9.28/5.65 | 17.34/7.41 | 27.22/**8.89** |
| *RED-Rec*-1.1B-LLaMA | 9.46/5.49 | 18.63/7.03 | 29.88/8.45 |
| *RED-Rec*-1.5B-Qwen2.5 | **9.98/5.88** | **19.98/7.88** | **32.62**/8.78 |

Table 6: Comparison on the Amazon Cloth-Sport cross-scenario setting. Metrics for Cloth are shown on the left, and for Sport on the right.

| Method | HR$_{10}$ (Cloth) | NDCG$_{10}$ (Cloth) | HR$_{10}$ (Sport) | NDCG$_{10}$ (Sport) |
|---|---|---|---|---|
| SASRec* | 70.57 | 0.6543 | 59.64 | 0.4900 |
| Bert4Rec* | 65.31 | 0.5720 | 53.50 | 0.4514 |
| AMID* | 74.75 | 0.6814 | 63.77 | 0.5867 |
| STAR | 62.36 | 0.5243 | 51.22 | 0.4724 |
| M2M | 65.27 | 0.5362 | 52.08 | 0.4626 |
| LLM4CDSR* | 80.18 | 0.7316 | 70.46 | 0.6312 |
| *RED-Rec*-1.1B-LLaMA | 76.28 | 0.6821 | 66.28 | 0.5824 |
| *RED-Rec*-1.5B-Qwen2.5 | 78.25 | 0.7214 | 71.12 | 0.6318 |

## C  DATASET

We compare our dataset with other existing datasests or benchmarks from UGC platforms in Table 7.

An item in the proposed dataset is like:

Listing 1: Example of an item in the training dataaset.

```
{
  "user_id": "xxxx",
  "data": {
    "homefeed_item_lastn": [
      {
        "duration": 28,
        "is_click": 1,
        "is_click_profile": 0,
        "is_collect": 0,
        "is_comment": 0,
        "is_follow": 0,
        "is_hide": 0,
        "is_like": 0,
        "is_nns": 0,
        "is_pagetime": 1,
        "is_read_comment": 1,
        "is_share": 0,
        "is_videoend": 0,
        "item_id": "684a48440000000023014319",
        "page_key": 0,
        "timestamp": 1749771247,
        "type": "note"
      },
      {
        "duration": 17,
        "is_click": 1,
        "is_click_profile": 0,
        "is_collect": 0,
        ...
        "is_pagetime": 1,
        "is_read_comment": 0,
        "is_share": 0,
        "is_videoend": 0,
        "item_id": "684aa0720000000021003dbe",
        "page_key": 0,
```

Table 7: A brief comparison of public-released datasets and the training dataset *RED-Rec* used.

| Property | Amazon | JD Search | KuaiSAR | Qilin | *RED-Rec*(dataset) |
|---|---|---|---|---|---|
| Users | 192.4k | 173.8k | 25.8k | 15.5k | 1.0m |
| Items | 63.0k | 12.9m | 6.9m | 2.0m | 300.6m |
| Queries | 3.2k | 171.7k | 453.7k | 571.9k | search items |
| Actions | 1.7m | 26.7m | 19.7m | 2.5m | 683.2m |
| Content | text/image | text | text/video | text/image/video | text/image/video |
| Scenario | Rec | Search | Search+Rec | Search+Rec | Search+Rec+Ads |

```
        "timestamp": 1749732355,
        "type": "note"
      }
      // ...
    ],
    "ads_item_lastn": [ ... ],
    "search_item_lastn": [ ... ]
  }
}
```

where

- **user_id**: Unique identifier for the user, e.g., `xxxx`.
- **data**:
  - **homefeed_item_lastn**: An array of objects representing the last $n$ items from the user's home feed. Each object contains:
    * `duration`: Viewing duration (in seconds).
    * `is_click`: Whether the item was clicked (1) or not (0).
    * `is_click_profile`: Whether the user's profile was clicked (1 or 0).
    * `is_collect`: Whether the item was collected or saved (1 or 0).
    * `is_comment`: Whether the item was commented on (1 or 0).
    * `is_follow`: Whether the user followed from this item (1 or 0).
    * `is_hide`: Whether the item was hidden (1 or 0).
    * `is_like`: Whether the item was liked (1 or 0).
    * `is_message`: Whether the author of the message was messaged (1 or 0).
    * `is_pagetime`: Whether the page time event was triggered (1 or 0).
    * `is_read_comment`: Whether comments were read (1 or 0).
    * `is_share`: Whether the item was shared (1 or 0).
    * `is_videoend`: Whether a video was watched until the end (1 or 0).
    * `item_id`: Identifier for the content item.
    * `page_key`: Page identifier.
    * `timestamp`: Timestamp of the interaction.
    * `type`: Type of item, e.g., `note`.
  - **ads_item_lastn**: Array of the last $n$ interacted advertisement items (`item_id`, `duration`, etc.).
  - **search_item_lastn**: Array of the last $n$ search items with similar structure.

Privacy protection is paramount in our dataset construction, implemented through multiple complementary techniques to safeguard user confidentiality. We employ comprehensive data anonymization by replacing real user with cryptographically secure hashes, ensuring unlinkability to original entities. We also add a consistent bias to the engagement timestamp. We retain only essential behavioral signals required for recommendation research while removing potentially identifying metadata such as device information, location data, and detailed content descriptors, thereby creating a privacy-preserving dataset that enables recommendation system research without compromising user privacy.

We focus exclusively on active platform users who demonstrate substantial engagement patterns: users must have at least 30 valid clicks in the homefeed scenario and 5 valid clicks in the advertisement scenario, where a click is considered valid only if the associated viewing duration exceeds 5 seconds.

# D METRICS

In this work, we focus on three widely adopted metrics: Hit Ratio (HR), Normalized Discounted Cumulative Gain (NDCG), and Mean Reciprocal Rank (MRR). In recommender systems and information retrieval, model performance is typically assessed by ranking-based evaluation metrics that reflect both the accuracy and the ordering of recommendations. These metrics are evaluated at various ranking cutoffs $K$ (e.g., $K = 10, 100, 1000$) to provide a comprehensive view of retrieval quality across different user engagement depths.

**Hit Ratio (HR)**  Hit Ratio (HR@K) measures the proportion of test cases in which at least one relevant item, usually the ground-truth item, is found within the top-$K$ positions of the ranked recommendation list. Formally, for a set of $N$ users (or queries), it is defined as:

$$\text{HR@}K = \frac{1}{N} \sum_{i=1}^{N} \mathbb{I}(\text{rank}_i \leqslant K), \tag{4}$$

where $\text{rank}_i$ denotes the position (starting from 1) at which the ground-truth item for the $i$-th user occurs in the predicted ranking, and $\mathbb{I}(\cdot)$ is the indicator function. HR is equivalent to recall@K in the case of a single relevant item per query.

HR@K is intuitive and interpretable, indicating the likelihood that a user's desired item appears among the top-$K$ recommendations. However, it does not reward higher placements within the top-$K$ and disregards the relative ranking among recommended items.

**Normalized Discounted Cumulative Gain (NDCG)**  Normalized Discounted Cumulative Gain (NDCG@K) extends HR@K by accounting for the position of relevant items, rewarding items that are ranked higher in the recommended list. For each test case, DCG is computed as:

$$\text{DCG@}K = \sum_{j=1}^{K} \frac{\text{rel}_{ij}}{\log_2(j+1)}, \tag{5}$$

where $\text{rel}_{ij}$ is the relevance label (typically 1 for the ground-truth item and 0 otherwise) for the $j$-th item in the ranked list for user $i$. The DCG is then normalized by the ideal DCG (IDCG), i.e., the maximum possible DCG for that user, to yield:

$$\text{NDCG@}K = \frac{1}{N} \sum_{i=1}^{N} \frac{\text{DCG}_i\text{@}K}{\text{IDCG}_i\text{@}K} \tag{6}$$

NDCG@K captures both the relevance and ranking quality, penalizing relevant items that appear lower in the ranking. It is especially useful in scenarios with multiple relevant items per user or graded relevance.

**Mean Reciprocal Rank (MRR)**  Mean Reciprocal Rank (MRR@K) evaluates how highly the first relevant item is ranked, and is defined as:

$$\text{MRR@}K = \frac{1}{N} \sum_{i=1}^{N} \frac{1}{\text{rank}_i}, \tag{7}$$

where $\text{rank}_i$ is the position of the first relevant item in the recommended list for user $i$, and set to infinity (i.e., reciprocal rank is 0) if no relevant items are found in the top-$K$. MRR@K emphasizes early precision, heavily rewarding algorithms that surface the relevant item at or near the top. Its sensitivity to the first relevant item's position makes it particularly apt for settings prioritizing immediate relevance (e.g., question answering, search).

**Evaluation Protocols and Cutoff Values**  In our work, all metrics above are computed at different cutoff values $K$ to approximate various user scenarios (e.g., users interacting with the top 10 or top 100 items). These are denoted as HR@$K$, NDCG@$K$, and MRR@$K$, for various $K$ (e.g., $K = 10, 100, 1000$). For interpretability and easier comparison, MRR is often multiplied by 100 and reported as $\text{MRR}_{*100}$. These metrics are computed under a leave-one-out or leave-many-out evaluation: for each user, one or more ground-truth relevant items are held out (used as positives), and the ranking is judged over a candidate pool comprising these positives and many sampled negatives.

# E  IMPLEMENTATION DETAILS

## E.1  MODEL IMPLEMENTATION

We provide additional implementation details of the proposed *RED-Rec*.

**Item Encoder.** The item encoder is designed to construct robust content representations, leveraging a pretrained LLM as its foundation. Textual information related to each item—including titles and descriptions—is concatenated, tokenized, and prepended with a designated special token to sharpen the representation focus. This sequence is then passed through the LLM encoder, producing dense semantic embeddings for each item. Specifically, we extract the embedding corresponding to the special token. The resulting embedding's dimension matches the model's hidden size; for instance, 1536 for LLaMA2-1.3B and 3584 for Qwen-7B.

For multimodal input, there are essentially two primary approaches. The first involves utilizing an individual vision encoder such as ViT(Dosovitskiy et al., 2020) like LLaVA(Liu et al., 2023), to extract visual tokens, which are then projected into the language embedding space. The second approach directly leverages vision-language models (VLMs) such as Qwen-VL, which jointly process visual and textual inputs within a unified architecture. In our work, we primarily adopt the first approach based on considerations of model size and efficiency for online serving.

**User Encoder.** User representation learning is managed via hierarchical interest modeling over long interaction histories. User interaction sequences are first encoded using the item encoder, resulting in contextualized item embeddings. These are then organized and refined by the proposed mixer module that captures temporal and sequential dependencies. The enhanced representations are subsequently fed into a disentangled multi-interest learning module, which goes beyond conventional single-vector user profiles by learning multiple independent embeddings—each attending to a distinct facet of user intent.

Training supervision extends past traditional next-item prediction, encompassing all interactions within a lookahead window to better reflect realistic browsing patterns. To achieve this, we apply cosine similarity clustering to partition target items based on behavioral signals, followed by Hungarian algorithm matching to associate each cluster centroid with its corresponding interest vector. A contrastive loss function drives the specialization of each embedding, ensuring broad coverage and effective disambiguation of diverse user preferences across multiple interest groups. Complete implementation details are available in our supplementary code repository.

To model user interests in a disentangled manner, we introduce learnable queries that capture refined, distinct interests according to three key principles: sufficient supervision for each query, minimal overlap in interest coverage, and coherent optimization directions. Given refined interest embeddings $\{\mathbf{r}_1, \ldots, \mathbf{r}_s\}$ and positive samples $\{\mathbf{t}_1, \ldots, \mathbf{t}_w\}$ from the target window, we cluster the positive samples into $s$ groups using cosine similarity and then match cluster centroids to interest embeddings via the Hungarian algorithm to maximize pairwise similarity. The contrastive loss is applied only to these matched pairs. The contrastive loss $\mathcal{L}_{\text{NCE}}$ is defined as:

$$\mathcal{L}_{\text{NCE}}(t, r) = -\log \frac{e^{\text{sim}(t,r)/\tau}}{e^{\text{sim}(t,r)/\tau} + \sum_{i=1}^{m} e^{\text{sim}(r,e_i)/\tau}} \tag{8}$$

where $m$ is the number of negative samples, $e_i$ is the $i$th negative sample embedding, and sim denotes cosine similarity.

This design enables adaptive learning: queries naturally specialize for users with diverse interests and converge for users whose preferences are more focused. nterest users while naturally converging for users with focused preferences.

## E.2 Online Deployment Optimization

In this section, we describe our practical optimizations at both the item and user sides, ensuring efficiency, scalability, and robustness in a real-world environment.

**Item-side** To enable fast candidate retrieval, we precompute and cache embeddings for all active items on the platform using our item encoder. Existing items are retraced in large batches every 7 days utilizing 64 H800 GPUs, while new items are indexed daily with 5 L20 cards on the item serving side. The resulting item embeddings are stored in a high-throughput, scalable key-value (KV) database for real-time lookup during serving. Embedding updates are performed asynchronously, ensuring the retrieval pipeline remains responsive even during refresh cycles. This caching strategy significantly reduces inference latency, as the model does not need to encode each item on-the-fly. Furthermore, we observe that the top 20% of items account for 99% of user clicks, and leveraging the KV cache effectively eliminates redundant computation for these high-frequency items.

**User-side** For user representation, we maintain a rolling window capturing each user's most recent behaviors on the platform. Upon each user request, the backend retrieves the corresponding item embeddings for these behaviors directly from the item KV cache. These cached embeddings are then fed into the user-side encoder, which leverages the LLM model to generate up-to-date user representations within strict latency constraints. To further minimize latency, user-side encoding is optimized through efficient batching, grouping requests to maximize GPU throughput during sequential modeling. We also implement intermediate representation caching: for highly active users, we cache and incrementally update their user embeddings as new interactions occur, recomputing only when substantial behavioral changes are detected. The user-side service uses 20 L20 cards.

**Model Compression and Quantization** To accommodate the deployment of large-scale parameters, we employ model quantization and compression techniques to ensure cost-effective and efficient inference. On the user side, we utilize bf16 quantization to accelerate computation and reduce memory usage, while on the item side, embeddings are maintained with 6 decimal places to further minimize bandit pressure and optimize cache efficiency. We use ONNX Runtime[3] for further acceleration, and we specifically precompile intensive operators such as multi-head attention and layer normalization for the LLM components to further minimize latency during online inference. These strategies decrease memory footprint and lower serving latency, particularly for user-side computation. All serving nodes operate statelessly, relying on distributed caches and databases for both embedding retrieval and user state management. This design is for seamless horizontal scaling. Additionally, if a user disables an item they have posted—rendering it invalid—the system automatically pads the sequence with empty content to ensure consistent input structure and model stability.

## F Further Experiments

### F.1 Pretraining Validation

Our first set of experiments investigates the effect of varying the backbone LLMs for the item and user encoders. Specifically, we explore the following configurations: (1) using different pretrained LLMs for item and user encoders, (2) training one or both encoders from scratch instead of initializing from a pretrained model, and (3) freezing the item encoder during training. The detailed results are summarized in Table 8.

Across all settings, we observe that using exactly the same pretrained LLM for both item and user encoders and fine-tuning them jointly yields the best performance. In contrast, utilizing mismatched

---

[3] https://github.com/microsoft/onnxruntime

Table 8: Ablation results for different combinations of item and user LLMs and training strategies.

| Scenario | Configuration | HR/NDCG$_{10}$ | HR/NDCG$_{100}$ | HR/NDCG$_{1k}$ | MRR$_{*100}$ |
|---|---|---|---|---|---|
| Homefeed | *RED-Rec* (2 * Qwen) | 2.31/0.68 | 12.59/1.88 | 31.94/3.86 | 1.27 |
| | Item LLM from scratch | 0.00/0.00 | 0.00/0.00 | 0.03/0.01 | 0.00 |
| | User LLM from scratch | 0.00/0.00 | 0.03/0.01 | 1.32/0.21 | 0.01 |
| | Item LLM frozen | 1.27/0.36 | 5.51/0.77 | 11.37/1.02 | 0.37 |
| | User LLM frozen | 1.78/0.44 | 10.47/1.02 | 23.06/1.48 | 1.01 |
| Homefeed + Ads | *RED-Rec* (2 * Qwen) | 4.36/1.31 | 18.32/3.27 | 42.61/5.02 | 2.11 |
| | Item LLM from scratch | 0.00/0.00 | 0.00/0.00 | 0.08/0.04 | 0.01 |
| | User LLM from scratch | 0.00/0.00 | 0.00/0.00 | 1.01/0.07 | 0.03 |
| | Item LLM frozen | 1.49/0.41 | 9.49/1.31 | 19.29/1.52 | 0.76 |
| | User LLM frozen | 2.57/1.01 | 13.72/1.98 | 29.72/1.88 | 3.28 |
| Homefeed + Ads | *RED-Rec* (2 * Qwen) | 4.36/1.31 | 18.32/3.27 | 42.61/5.02 | 2.11 |
| | *RED-Rec*-CoT (2 * Qwen) | 4.46/1.35 | 18.78/3.60 | 44.61/5.01 | 2.15 |

encoders, initializing from scratch, or freezing either encoder all result in significant drops in overall accuracy. This suggests that consistent representation spaces and co-adaptation between the two encoders are crucial for optimal model performance.

## F.2 CoT Validation

We explore explainable recommendations based on Chain-of-Thought (CoT)-based (Wei et al., 2022) explanations for the input layer in multi-scenario setting. In this experiment, we introduce a Chain-of-Thought (CoT) auxiliary loss: beyond learning discriminative user and item encoders, we encourage explainable multi-scenario reasoning by forcing the user model to generate natural language rationales for each action:

$$\mathcal{L}_{\text{CoT}} = -\sum_{u \in \mathcal{U}} \sum_{t=1}^{|S_u|} \sum_{\ell=1}^{L_t} \log p_\phi(r_{t,\ell} \mid r_{t<\ell}, \mathbf{z}_{u,t}), \quad (9)$$

where $p_\phi()$ denotes the probability, computed by a learnable language model head parameterized by $\phi$, of generating the $\ell$-th token $r_{t,\ell}$ of the rationale conditioned on the previous tokens $r_{t<\ell}$ and the contextualized user embedding $\mathbf{z}_{u,t}$ at interaction $t$. The overall training loss is then $\mathcal{L}_{\text{total}} = \mathcal{L}_{\text{NCE}} + \lambda_{\text{CoT}}\mathcal{L}_{\text{CoT}}$.

We use GPT 4.1 to generate CoT explanations. An example of the generated CoT explanantion is like:

*"The user browsed multiple articles related to Switzerland on the homepage, such as "Do you dare to guess how many days of sunshine in Switzerland?" and "What to wear for a trip to Switzerland next week?" This indicates a clear interest in Switzerland. While previously recommended ads included those related to travel, they were not specifically targeted at Switzerland.*

*Therefore, we recommend to the user the targeted ad "Personal tested and useful! The ultimate transportation ticket map tool for traveling in Switzerland!", as well as other ads related to traveling in Switzerland, such as "Countdown to opening! The four legendary theme parks of Fiesch First Mountain" and "Interlaken sledding premium tips — Save 400 RMB instantly".*

The CoT explanation module is particularly well-suited to the multi-scenario recommendation setting. By generating step-by-step rationales that account for user behaviors across different scenarios or domains, the model can provide contextually accurate and human-understandable justifications for its recommendations. This improves both transparency and user trust, crucial for scenario-aware systems. However, we observe that applying the CoT-based approach to large-scale datasets introduces significant challenges. The requirement to generate context-dependent rationales for every user interaction leads to substantially increased computational and memory costs. Given these limitations, we restrict our experiments to small-scale testings. The detailed results are summarized in Table 8. Including CoT data in training has led to certain improvements, but it does not outperform the pretrained model.

