# OpenReview forum: "Cross-Scenario Unified Modeling of User Interests at Billion Scale"
_ICLR.cc/2026/Conference — ICLR 2026 Conference Withdrawn Submission_

### Official Review · Reviewer_5t77 · 2025-10-25

**Soundness:** 3
**Presentation:** 3
**Contribution:** 2
**Rating:** 4
**Confidence:** 5

**Summary:**

This paper proposes RED-Rec, an LLM-enhanced multi-scenario sequential recommendation framework. The authors also collect a million-scale multi-scenario sequential dataset from a UGC platform.

**Strengths:**

S1: This paper studies an important topic on multi-scenario sequential recommendation.

S2: This paper is clearly written and easy to read.

S3: This paper provides a new million-scale industrial dataset covering different scenarios and channels.

**Weaknesses:**

1. The authors neglect an important paper [1] on the same topic, i.e., cross-domain sequential recommendation. The proposed method LLM4CDSR of the reference paper is an important baseline method which need to be discussed.

2. From my perspective, the potential impact and contribution of this paper is not significant. First, the necessity of introducing an LLM-enriched framework is not fully illustrated. Second, the previous work HLLM has already proposed a two-tower framework with a user tower and an item tower both adopting LLM architecture. Upon that, RED-Rec purely introduce multimodal information, 2-D positional encoding, and scenario-aware queries.

3. The experiments are only conducted on industrial datasets without validation on public representative datasets like Amazon.

4. The presentation needs improving. For example, Figure 3 is cluttered and unclear. Besides, the Appendix is missing and Appendix D is not mentioned.

[1] Bridge the Domains: Large Language Models Enhanced Cross-domain Sequential Recommendation, SIGIR 2025

**Questions:**

1. The Appendix mentioned is not included in the manuscript and there is no supplementary material.

2. In the experiment part, is it fair to compare with the pre-trained model variants (i.e., '-pt'), which are trained on a large-scale online dataset? This is because such implementation introduces additional information.

3. Some details of experiments are missing. For example, does the implemented HLLM use multimodal information?

4. Please refer to the third weakness. Will the authors include experimental results on public multi-scenario dataset? Generally this paper would be a better fit for the industry track of a data mining conference (e.g., ADS track of KDD).

---

> ### Author Response · Authors · 2025-11-21
>
> Dear Reviewer 5t77,
>
> Thank you very much for your valuable feedback. Below, we address each of your points:
>
> > The authors neglect an important paper [1] on the same topic, i.e., cross-domain sequential recommendation.
>
> Thank you for highlighting this important work. We have updated our paper to include a comparative discussion of this important work LLM4CDSR in Sec 2 and also Sec Supp B.
>
> > First, the necessity of introducing an LLM-enriched framework is not fully illustrated. Second, the previous work HLLM has already proposed a two-tower framework with a user tower and an item tower both adopting LLM architecture.
>
> We respectfully disagree with the view, and would like to elaborate on our key contributions. The two-tower framework is indeed a classic and widely adopted paradigm in recommendation systems. Replacing MLP-based towers with LLM-based architectures is a natural and emerging trend in the community, as exemplified by recent works such as HLLM. However, our work differs in important ways: our main focus is not simply on adopting LLMs for user and item modeling, but on addressing the practical challenges of deploying such architectures in large-scale, real-world scenarios. Specifically, we propose a comprehensive framework to tackle highly unbalanced, long-tailed, and complex industrial recommendation tasks. We introduce the overall framework, the mixing and  querying policy not as incremental features, but as critical components to enable the LLM-based two-tower structure to perform effectively at industrial scale. Our offline and online experiments demonstrate the effectiveness of our approach in extremely complex and large-scale environments—settings that, to our knowledge, have not been thoroughly studied in prior works.
>
> > The experiments are only conducted on industrial datasets without validation on public representative datasets like Amazon.
>
> Thank you for your valuable suggestion. In response, we have added experiments on the Amazon dataset and revised the paper accordingly. We further test and compare RED-Rec in an cross-scenario setting, where we choose the Cloth-Sport sub-categories of the Amazon dataset. We kindly refer you to Supp Sec B for the new results and detailed analysis. Furthermore, to facilitate fair comparison and reproducibility, we have released our training and test data to the community. We appreciate your feedback for helping us strengthen our work.
>
> > Figure 3 is cluttered and unclear. Besides, the Appendix is missing.
>
> We have fixed all broken links and now provide the full code, dataset, and checkpoints anonymously for replication (https://anonymous.4open.science/r/RedSeqRec-ano-4158). We kindly refer you to the revised paper for further details. We apologize for the absence of these materials in the initial submission. We hope it is not too late, and hope that these additions will help improve the understanding and reproducibility of our work.
>
> > In the experiment part, is it fair to compare with the pre-trained model variants (i.e., '-pt'), which are trained on a large-scale online dataset? This is because such implementation introduces additional information.
>
> Thank you for raising this important point. The pre-trained model variants are not trained on a large-scale online dataset, but rather on additional user sequences from the same source as our primary experiments. Our intention here is to demonstrate that RED-Rec can effectively handle a greater volume of data and that its performance continues to scale as more user information is provided. To ensure fairness, we have also included models trained strictly on the same training dataset, allowing for direct and equitable comparisons. We hope this clarifies our experimental design.
>
> > Does the implemented HLLM use multimodal information?
>
> Thank you for your attention to the experimental details. To clarify, HLLM is officially released as a pure text-based LLM framework, and our implementation follows this standard setup. Although the HLLM paper compares its approach to vision-inclusive baselines such as SASRec (which utilizes ViT as the image encoder for item representation), the reported results show no significant improvement with the addition of multimodal information. For this reason, we did not include multimodal extensions in our HLLM implementation. We have also reported RED-Rec results without multimodal data for a fair comparison.
>
> > Will the authors include experimental results on the public multi-scenario dataset?
>
> Yes. To address this point, we have added two sets of new experiments. The first is conducted on Amazon Books Reviews. The second follows the setup in LLM4CDSR and utilizes the Amazon Cloth-Sport dataset, which is designed for cross-scenario reasoning tasks. We compare our Qwen2.5-based model with baseline results, and the outcomes are presented in Supplementary Section B. We kindly refer you to the revised paper for the experiment resuls and the discussion.

---

### Official Review · Reviewer_Z4KA · 2025-10-29

**Soundness:** 3
**Presentation:** 3
**Contribution:** 3
**Rating:** 4
**Confidence:** 4

**Summary:**

The paper presents RED-Rec, a large-scale LLM-enhanced hierarchical rec framework designed to unify user interest modeling across multiple scenarios or domains. Traditional recsys are typically scenario-specific and fail to capture cross-scenario dependencies, leading to fragmented user understanding. To address this, RED-Rec introduces a two-tower LLM-powered architecture that integrates textual and visual features through multimodal encoders and employs a 2-D dense mixing policy to fuse behavioral signals along temporal and scenario dimensions. It also incorporates scenario-aware queries to express fine-grained user intents. The authors construct a million-scale multi-scenario sequential dataset from a large UGC platform and validate RED-Rec through both offline benchmarking and online A/B tests on hundreds of millions of users. Results show significant performance gains over strong baselines, demonstrating the feasibility of unified cross-scenario recommendation at an industrial scale.

**Strengths:**

1. The paper is clearly written and logically well-structured.
2. Introducing sequential modeling into the multi-scenario domain is indeed a very interesting problem.

**Weaknesses:**

1. The related work section is not comprehensive, and many studies on multi-scenario recommendation have been overlooked, such as the work in this project [1].
2. In line 212, what does H_u represent?
3. How exactly is the model trained? The authors only mention in line 323 that NCE is used as the main objective, but the training procedure remains unclear to me.
4. Figure 4 refers to negative sampling multiple times, yet this is not explicitly described in the methodology section.
5. Where can the appendices mentioned in the text be found? I could not locate them.
6. Section 5.3 omits several classic multi-scenario recommendation algorithms, such as STAR [2] and M2M [3].

[1] https://github.com/Xiaopengli1/Scenario-Wise-Rec

[2] Sheng X R, Zhao L, Zhou G, et al. One model to serve all: Star topology adaptive recommender for multi-domain ctr prediction[C]//Proceedings of the 30th ACM International Conference on Information & Knowledge Management. 2021: 4104-4113.

[3] Zhang, Qianqian, et al. "Leaving no one behind: A multi-scenario multi-task meta learning approach for advertiser modeling." Proceedings of the Fifteenth ACM International Conference on Web Search and Data Mining. 2022.

**Questions:**

please refer to the weakness section

---

> ### Author Response · Authors · 2025-11-21
>
> Dear Reviewer Z4KA,
>
> Thank you very much for your valuable feedback. Below, we address each of your points:
>
> > The related work section is not comprehensive, and many studies on multi-scenario recommendation have been overlooked, such as the work in this project [1].
>
> Thank you for highlighting these important works. We have revised the related work part and included these works. We would be glad to incorporate more if you have additional references or resources to share.
>
> > In line 212, what does H_u represent?
>
> H_u denotes the user history sequences, which are sequences of items sorted in reverse chronological order. See line 312-317.
>
> > How exactly is the model trained?
>
> We thank the reviewer for this important question. To clarify, our model is trained end-to-end using a contrastive loss based on Noise Contrastive Estimation. We have now added Supp Sec E.2 in the supplementary material to provide a detailed and explicit description of the training process for the learning module, clearly explaining how the contrastive objective, clustering, and matching components are integrated, and also add more details in Sec 4.2.
>
> Briefly, positive samples within the observation window are used to learn refined interest representations in a disentangled manner. Specifically, we cluster the user’s positive samples, align these clusters with learnable query-based interest embeddings using the Hungarian algorithm, and then apply contrastive learning to encourage specialization and diversity among the interest embeddings. Our intuition is to ensure that each query receives targeted supervision and avoids overlapping interest coverage, enabling stable and effective learning of user preferences.
>
> > Figure 4 refers to negative sampling multiple times, yet this is not explicitly described in the methodology section.
>
> The negative sampling is quite simple. We randomly select unclicked samples as negative examples for the contrastive loss.
>
> > Where can the appendices mentioned in the text be found? I could not locate them.
>
> We have fixed all broken links and now provide the full code, dataset, and checkpoints anonymously for replication (https://anonymous.4open.science/r/RedSeqRec-ano-4158). We kindly refer you to the revised paper for further details. We apologize for the absence of these materials in the initial submission. We hope it is not too late, and hope that these additions will help improve the understanding and reproducibility of our work.
>
> > Section 5.3 omits several classic multi-scenario recommendation algorithms, such as STAR and M2M.
>
> Thank you for highlighting this important point. We have updated the relevant section to include a comparative discussion of these seminal algorithms in Sec Supp B, presenting not only results for STAR and M2M but also for other multi-scenario recommendation models suggested by reviewers. Specifically, we conducted experiments on the Amazon dataset, selecting “Clothing” and “Sports” as the multi-scenario recommendation set. Our results indicate that RED-Rec surpasses these baselines in most cases. It is worth noting that the performance of STAR and M2M is slightly different from what was reported in Scenario-Wise-Rec, as our evaluations were conducted on a different scenario splits.
>
> Furthermore, we would like to explain that RED-Rec and approaches such as STAR, M2M, and LLM4CDSR are designed from fundamentally different perspectives. Namely, academic datasets and benchmarks often prioritize interest generalization across disjoint interest groups (e.g., transitioning from “Clothing” to “Sports”) through dedicated architectural designs, whereas RED-Rec is intended for large-scale unified modeling across business domains, focusing on holistic user behavior modeling. From this perspective, our work aligns more closely with models such as HSTU and HLLM. Accordingly, we compare our approach with these baselines on the proposed large-scale industrial datasets. In response to your question, we have included comparisons with STAR and M2M on publicly available academic datasets.

---

### Official Review · Reviewer_w6NV · 2025-10-30

**Soundness:** 2
**Presentation:** 3
**Contribution:** 1
**Rating:** 2
**Confidence:** 4

**Summary:**

This paper focuses on the unified modeling of multi-scenario recommendation. The authors propose RED-Rec, a rather complex systematic modeling framework that aims to unify the modeling of various features across scenarios such as homefeed, ads, and search, mainly through semantic encoding and scenario-aware processing. The framework leverages LLM-based item and user encoders, along with a dense mixing policy to integrate multi-scenario signals. Experiments on an industrial dataset and online A/B tests suggest the method is useful and potentially highly effective in industrial settings. However, the approach is more of a comprehensive industrial implementation that integrates various existing technologies rather than introducing a clearly innovative model.

**Strengths:**

- The paper addresses a real-world and practical issue in large-scale multi-scenario recommendation systems.
- The proposed unified modeling approach is well-structured and demonstrates performance gains in industrial-scale deployment.
- Empirical validation includes both offline experiments and large-scale online A/B testing, providing strong practical evidence.
- The design effectively integrates multiple technologies, indicating high engineering value for industry applications.

**Weaknesses:**

- Limited novelty — the modeling method appears more like an aggregation of existing techniques (LLM-based encoders, multi-scenario mixing) into a unified industrial system rather than introducing fundamentally new algorithms.
- Missing comparisons with recent, relevant multi-scenario recommendation baselines (e.g., STAR, APG, AdaSparse, HierRec), which weakens the claim of state-of-the-art performance in this subfield.
- All appendix links are invalid, preventing reviewers from accessing potentially important supplementary definitions, dataset details, and experimental settings. This significantly impacts reproducibility.
- No anonymous, reproducible code or dataset provided at review time, despite the promise of releasing them upon acceptance. This limits the ability to verify implementation details.
- The evaluation leans heavily on proprietary industrial data; while practical, this limits independent verification and reduces the openness of the contribution.

**Questions:**

- The modeling method is more like an industrial implementation that integrates various technologies rather than an innovative research contribution.
- The paper belongs to the field of multi-scenario recommendation, but the experiments do not compare against recent multi-scenario recommendation methods such as STAR, APG, AdaSparse, HierRec, etc., only against some general recommendation methods, which is inappropriate.
- Although the paper states that the code will be made open-source upon acceptance, no anonymous and reproducible code is currently provided for review.
- All links to the appendices are invalid, preventing reviewers from accessing any of the appendix contents.

---

> ### Author Response · Authors · 2025-11-21
>
> Dear Reviewer w6NVj,
>
> Thank you very much for your valuable feedback. Below, we address each of your points:
>
> > The modeling method appears more like an aggregation of existing techniques (LLM-based encoders, multi-scenario mixing) into a unified industrial system rather than introducing fundamentally new algorithms.
>
> We respectfully disagree with the characterization of our work as lacking in novelty, or being merely an aggregation of existing technologies. Our primary motivation is to address recommendation challenges in real-world, industrial-scale, multi-scenario environments—a setting rarely tackled with fully semantic, end-to-end frameworks in deployed practice. To the best of our knowledge, there is currently no prior work that has introduced or validated an end-to-end semantic-based recommendation architecture at this scale and complexity in production systems. We would also like to emphasize that novelty should not be solely defined by complex formulas or elaborate architectural blocks. Instead, we believe practical impact and clean, well-engineered solutions that demonstrate robust performance in real-world settings are equally important contributions.
>
> > Missing comparisons with recent, relevant multi-scenario recommendation baselines (e.g., STAR, APG, AdaSparse, HierRec)
>
> Thank you for highlighting these important multi-scenario recommendation models. We greatly agree that these baselines are important and have now included relevant citations and discussions in our revised paper, along with further experimental results comparing our model to multi-scenario recommendation baselines.
>
> Among these works you mentioned, AdaSparse and HierRec are more close to our works, while we thoroughly searched but were unable to find publicly available official training code implementations, which makes direct comparison challenging. In the original submission, to ensure reproducibility and fairness, we focused on comparisons with widely-used open-source baselines such as HSTU and HLLM. In revision, additionally, following suggestions from Reviewer 5t77, we included experiments on public datasets and compared our model against multi-scenario baselines like LLM4CDSR and AMID. We test and compare \ac{method} in an cross-scenario setting, where we choose the Cloth-Sport sub-categories of the Amazon dataset following previous works. We kindly refer you to Section Supp B for further details and results.
>
> We appreciate your understanding and would be glad to incorporate direct comparisons with these works if you have additional references or implementation resources to share.
>
> > All appendix links are invalid ... No anonymous, reproducible code or dataset provided at review time.
>
> We have fixed all broken links and now provide the full code, dataset, and checkpoints anonymously for replication (https://anonymous.4open.science/r/RedSeqRec-ano-4158). We kindly refer you to the revised paper for further details. We apologize for the absence of these materials in the initial submission. We hope it is not too late, and hope that these additions will help improve the understanding and reproducibility of our work.
>
> > The evaluation leans heavily on proprietary industrial data; while practical, this limits independent verification and reduces the openness of the contribution.
>
> Thank you for highlighting the importance of independent verification and openness. We have now open-sourced our training and testing datasets, along with model checkpoints, to enhance reproducibility. In addition, we conducted further experiments on the widely-used, public Amazon dataset and report detailed performance comparisons with strong baselines in Section Supp B. These results demonstrate that our method consistently outperforms other approaches not only in industrial deployments but also on open academic datasets, broadening the applicability and openness of our contribution.
>
> For additional questions corresponding to these points, please refer to the above responses. Thank you again for your insightful suggestions.

---

### Official Review · Reviewer_NWcj · 2025-11-01

**Soundness:** 3
**Presentation:** 2
**Contribution:** 3
**Rating:** 6
**Confidence:** 2

**Summary:**

RED-Rec is designed to address the limitations of traditional RS that operate in isolated silos (e.g., separate models for feed, search, and ads). The authors argue that this siloed approach fails to capture a holistic understanding of user interests, which manifest across these different interaction contexts.

RED-Rec is thus proposed as a unified, user-centric framework tailored for billion-scale industrial deployments. Its architecture is an LLM-enhanced hierarchical two-tower model that learns comprehensive user and item representations by synthesizing behavioral signals from multiple scenario

key contributions include:
* LLM-powered encoder, which area two-tower structure where both the user and item encoders are powered by Large Language Models, enabling rich semantic representations from content and user histories
* 2-D dense mixing/qerying policy, which is designed to effectively fuse behavioral signals from different scenarios. It operates along two axes <scenario, time> to address data imbalances (e.g., more feed interactions than ad clicks) and capture cross-scenario user intent. It also uses "scenario-aware queries" to express fine-grained, context-specific user interests during serving. I think this second contribution is more interesting.

The framework was validated through extensive offline experiments and large-scale online A/B tests on a major UGC platform, where it showed substantial performance gains in both content recommendation and advertising

**Strengths:**

Originality:

"2-D dense mixing and querying policy" is a novel and specific technical contribution designed to handle the practical challenges of multi-scenario data. By fusing signals along both temporal and scenario axes, it explicitly addresses data imbalance (e.g., more feed interactions than ad clicks), ensuring that infrequent but valuable user signals are not lost.

Quality
* The paper's most significant strength is its validation through online A/B tests on a world-leading UGC platform supporting hundreds of millions of daily users.
* They contribute to the quality of future research by introducing a new, million-scale multi-scenario sequential dataset.

Clarity:  The paper is mostly clear-written.

Significance
* with the industrial a/b testing result, the work shows a high practical impact.

**Weaknesses:**

* The paper's title and abstract heavily emphasize the "LLM-enhanced" nature of the framework. However, the experimental section lacks a crucial baseline: a comparison against a similar architecture that uses a non-LLM encoder (e.g., a standard Transformer or GRU operating on ID embeddings). Without this comparison, it is difficult to quantify the actual performance gain attributable to the expensive LLM component versus the gains from the unified architecture itself. The current ablation study only explores variations of the RED-Rec model, not its core components against simpler alternatives. A non-LLM would clearly demonstrate the value added by the language model.

* The paper presents a sophisticated two-tower hierarchical architecture. While powerful, it also introduces significant engineering complexity. The paper mentions "system-level optimizations enabling stable, low-latency online deployment" but provides no details on what these optimizations are. This makes it difficult to assess the full cost and engineering effort required to make such a system viable in production. While some details may be proprietary, providing a high-level overview of the types of optimizations employed (e.g., model quantization, caching strategies, asynchronous embedding generation) would greatly improve the paper's practical value and reproducibility.

* Clarity: the paper is mostly clear though the implementation of "2-D dense mixing" policy is not fully detailed. The paper states that the Merge(·) function "deterministically fuses events - sorting by timestamp and concatenating per a fixed scenario order," but the clarity could be improved by providing a more explicit description of the merging logic, including the "fixed scenario order" and the rationale behind it.

* The paper's primary claim is that it advances "unified modeling" for multi-scenario recommendation. However, the baselines used in the experiments are primarily strong single-scenario sequential recommendation models, not models explicitly designed for the multi-scenario setting. The related work section mentions several such models, but they are not included in the comparison.

* The online A/B test results are a major strength, but the details provided are sparse. The authors should provide more context for the A/B test. Describing the control group and reporting on the trade-offs - for example, did the focus on ads have any negative impact on organic content engagement?

**Questions:**

See Weakness section above

---

> ### Author Response · Authors · 2025-11-21
>
> Dear Reviewer NWcj,
>
> Thank you very much for your valuable feedback. Below, we address each of your points:
>
> > The paper's title and abstract heavily emphasize the "LLM-enhanced" nature of the framework. However, the experimental section lacks a crucial baseline: a comparison against a similar architecture that uses a non-LLM encoder (e.g., a standard Transformer or GRU operating on ID embeddings)....A non-LLM would clearly demonstrate the value added by the language model.
>
> We fully agree on the importance of isolating the effect of the LLM component. We have included results for both the widely-used HSTU and DLRM models, each recognized in academia and industry as strong ID-based baselines—often outperforming classical Transformer- or GRU-based architectures on recommendation tasks that primarily rely on categorical features. For example, on the MovieLens dataset, GRU4Rec achieves an HR@10 of 0.2811 and NDCG@10 of 0.1648, while HSTU further improves these numbers to 0.3097 and 0.1720, respectively. Our experimental results demonstrate that, although DLRM and HSTU are highly competitive on standard ID-based inputs, semantic-based methods (such as ours and HLLM) achieve even greater improvements when the corpus contains rich semantic information. This suggests that, at least in our settings, the semantic modeling capabilities introduced by LLMs offer clear benefits beyond those of purely ID-based approaches like DLRM and HSTU.
>
> > The paper mentions "system-level optimizations enabling stable, low-latency online deployment" but provides no details on what these optimizations are.
>
> We apologize for the earlier lack of detail. We have included a new section in supplementary materials that give an overview of our system-level optimizations. Please refer to Sec Supp E.2 for more details.
>
> > Clarity of “2-D dense mixing” Policy
>
> Our Merge(·) function fuses user interaction events from multiple scenarios (e.g., homefeed, ads, search) according to two axes: temporal order and scenario quota. For the fixed scenario order, we first define a scenario priority order, e.g., [homefeed, ads, search], determined based on their relative frequencies and business value in real-world user behavior. This order is set to ensure that less frequent but potentially more informative scenarios like ads or search are not overwhelmed by high-volume homefeed actions. For each user, we sample the most recent events from each scenario, where the number is scenario-specific and tuned as a hyperparameter to balance data representation. Then all selected events are merged into a single sequence, tagged by scenario. The merged events are then sorted globally by timestamp, ensuring chronological order irrespective of scenario. During concatenation, scenario tags are retained for positional encoding.
>
> This design achieves a balanced integration: 1. Temporal sorting guarantees modeling of cross-scenario temporal dependencies. 2. Fixed scenario quotas prevent dominant scenarios (e.g., homefeed) from masking rare signals (e.g., search or ads) in user history, which is critical for effective multi-scenario interest modeling.
>
> > However, the baselines used in the experiments are primarily strong single-scenario sequential recommendation models, not models explicitly designed for the multi-scenario setting. The related work section mentions several such models, but they are not included in the comparison.
>
> Thank you for your valuable suggestion. We acknowledge this limitation and have addressed it by adding additional results in Section Supp B, where we demonstrate that our model also outperforms multi-scenario baselines in most cases. We would also like to note that, while several academic models for multi-scenario recommendation are discussed in the related work section, most of these are designed specifically for research datasets and lack open-source implementations that are suitable for industrial-scale scenarios. To ensure fair and reproducible comparisons, our main experiments focus on widely adopted and actively maintained baselines that are commonly used in production or large-scale settings. Furthermore, for all models—including single-scenario baselines—we treat the item sequence as a unified sequence. This aligns with the methodological setting of our work and ensures a consistent evaluation protocol across all approaches.
>
> > The online A/B test results are a major strength, but the details provided are sparse. The authors should provide more context for the A/B test.
>
> Thank you for your valuable suggestion. We have included more details in Sec 5.6. We kindly refer you to check the revised paper for more details.

---

### Author Response · Authors · 2025-11-21
**Overall Response**

Dear AC and Reviewers,

We are very grateful to the committee and reviewers for their valuable feedback. We have revised our paper accordingly. Specifically, we have fixed all broken links and provided the full code, dataset, and checkpoints anonymously for replication (https://anonymous.4open.science/r/RedSeqRec-ano-4158). We apologize for the lack of certain materials in the initial submission, and we hope these additions will help improve the understanding and reproducibility of our work.

Our main revisions include incorporating related references and similar work into the discussion, adding more technical details about our method and online A/B testing, and presenting additional results on publicly available datasets with comparisons to existing methods.
We sincerely thank the reviewers again for their valuable advice. We kindly refer the reviewers to our revised paper and the provided code and datasets for further details.

---

### Note · Authors · 2026-01-26

I have read and agree with the venue's withdrawal policy on behalf of myself and my co-authors.

---

### Meta-Review · Area_Chair_2UdD · 2026-01-08

**Summary:**

This paper introduces RED-Rec, an LLM-enhanced hierarchical recommender system for unified multi-scenario modeling at billion-scale, validated with large-scale online A/B tests. Reviewers acknowledged the strong industrial relevance, large-scale deployment, and the proposed 2-D dense mixing policy as interesting. However, significant concerns were raised about attribution of gains to LLMs, missing or late baselines (non-LLM and multi-scenario), limited methodological clarity, and sparse details on system optimizations and A/B testing protocols. Despite impressive claims, reviewers were not fully convinced that the scientific contribution was clearly isolated from engineering scale effects.

**Reviewer Concerns:**

The authors substantially improved reproducibility by releasing code and datasets and added new baselines and explanations. While this addressed some clarity issues, key concerns persist. In particular, the value added by the LLM component remains difficult to disentangle from architectural and data advantages. Comparisons to established multi-scenario academic models are still limited, and online A/B testing details remain high-level. Given the borderline nature of the initial scores and remaining uncertainty about novelty versus engineering complexity, confidence did not rise sufficiently.

**Reviewer Scores:**

Reviewer NWcj (6): Likely remains 6 (borderline).
Other reviewers (~4): Likely remain 4 despite added material.
Overall consensus remains mixed and below a clear accept.

---

### Decision · Program_Chairs · 2026-01-26

Reject